# Comprehensive Analysis Identifies PKP3 Overexpression in Pancreatic Cancer Related to Unfavorable Prognosis

**DOI:** 10.3390/biomedicines11092472

**Published:** 2023-09-06

**Authors:** Yan Du, Shuang Hou, Zhou Chen, Wancheng Li, Xin Li, Wence Zhou

**Affiliations:** 1The Second School of Clinical Medicine, Lanzhou University, Lanzhou 730030, China; 2Department of General Surgery, The First Hospital of Lanzhou University, Lanzhou 730030, China; 3Department of General Surgery, Lanzhou University Second Hospital, Lanzhou 730030, China

**Keywords:** PKP3, pancreatic cancer, prognosis, immune infiltration, biomarker

## Abstract

Plakophilin 3 (PKP3) affects cell signal transduction and cell adhesion and performs a crucial function in tumorigenesis. The current investigation evaluated the predictive significance and underlying processes of PKP3 within pancreatic cancer (PC) tissues. The assessment of differences in PKP3 expression was conducted through an analysis of RNA-seq data acquired from The Cancer Genome Atlas (TCGA) and Gene Expression Omnibus (GEO) databases. Additionally, clinical samples were collected to validate the findings. The predictive significance of PKP3 was investigated by analyzing survival data derived from TCGA and clinical specimens. PKP3′s biological function was assessed via phenotypic experiments after the suppression of PKP3 expression within PC cells. Functional enrichment analysis, encompassing KEGG, GO, and GSEA, was employed to assess the underlying mechanism of PKP3. Immune infiltration analysis was conducted in the present investigation to determine the association between PKP3 and tumor-infiltrating immune cells (TICs). In PC tissues, PKP3 expression was abnormally upregulated and correlated with a negative prognosis in individuals with PC. PKP3 can promote the progression, migration, and invasive capacity of PC cells and is relevant to the regulation of the PI3K–Akt and MAPK signaling pathways. Immune infiltration analysis demonstrated that PKP3 impeded CD8+ T-cell infiltration and immune cytokine expression within the tumor microenvironment. The PKP3 protein was identified as a prospective independent predictive indicator and represents a viable approach for immunotherapy in the context of PC. PKP3 may impact prognosis by broadly inhibiting immune cell infiltration and promoting the activation of tumor-associated signaling pathways.

## 1. Introduction

PC is a malignancy correlated with elevated fatality rates. The prevalence of PC has risen by approximately 0.4% annually alongside the expansion of public health epidemics and the growth of the worldwide population [1]. PC’s five-year overall survival rate, which was previously estimated at approximately 10%, has improved in recent years. Nevertheless, PC continues to exhibit the most unfavorable prognosis among common solid cancerous malignancies [2]. Due to the absence of initial clinical manifestations in PC, individuals with PC are frequently diagnosed at an advanced stage, with over 80% of patients being unsuitable for surgical removal [3]. Patients with unresectable PC require radiotherapy, chemotherapy, and immunotherapy. However, the presence of drug resistance in tumors, hyperplasia of connective tissue, and an immunosuppressive microenvironment pose significant challenges to the efficacy of systemic treatment. These factors enable the tumor to progress quickly and few patients to achieve complete pathological remission [4,5]. Hence, it is imperative to determine diagnostic biomarkers and develop novel therapeutic approaches for patients with PC in order to enhance their clinical outcomes.

Plakophilins (PKPs) belong to the armadillo-repeat protein family, which comprises the p120- and beta-catenin subfamilies. These subfamilies all contain tandem arrays of a 42-amino-acid repeat motif [6]. Cell–cell junctions perform a vital function in the formation and preservation of cytoarchitecture, including adherent junctions, desmosomes, tight junctions, and gap junctions. The PKP protein family is involved in regulating the process of desmosome adhesion between cells by interacting with several desmosomal components (desmocollins, desmogleins, and desmoplakin), which contribute to the maintenance of tissue integrity [7,8,9]. PKPs participate in signal transduction pathways that regulate diverse biological processes such as protein synthesis, development, proliferation, cell migration, and tumorigenesis [10,11]. PKP3 is the most prevalent member of the PKP subfamily and exhibits high levels of expression in desmosome-bearing cells, except in the desmosomes present between hepatocytes and cardiomyocytes [12]. PKP3 is a critical factor in preserving the structural and functional integrity of bridging granules by facilitating the recruitment of other bridging granule constituents to the cellular boundaries [13]. Dysregulation of PKP3 in tumor cells leads to altered desmosome size, reduced intercellular adhesion, and increased cell migration, further contributing to tumor progression [14]. Previous investigations have presented the contribution of PKP3 in the initiation of multiple tumors, such as colon, lung, prostate, ovarian, and breast malignancies [15,16,17,18,19,20].

Recent investigations have revealed that the deficiency of PKP3 in a mouse model of colon cancer is connected with the upregulation of Lipocalin2, leading to adenoma formation, rectal prolapse, and reduced survival [15]. PKP3 functions in the tumor microenvironment of prostate cancer, regulating invasion of the cell and development of the tumor through the MMP7 protein, and is associated with poor patient prognosis. PKP3 performs an essential function in the prostate cancer tumor microenvironment by regulating invasion of the cell and tumor formation through the MMP7 protein. PKP3 is also correlated with negative patient prognosis [16,17]. In non-small-cell lung carcinoma, PKP3 impaired CD8+ T-cell-dependent antitumor immunity in an immunoreactive mouse model by engaging in competing endogenous RNA mechanisms [18]. These studies suggest that the potential mechanisms of PKP3′s involvement in tumors may be related to metabolic regulation and immune modulation of the tumor microenvironment. However, the impact of PKP3 within different PCs and its correlation with the PC immune microenvironment have not yet been investigated. In this investigation, the differential expression, prognostic value, and immune infiltration correlation of PKP3 in PC were evaluated based on TCGA-PAAD and GEO-PAAD datasets. Then, we collected clinical samples to validate the diagnostic value and prognostic predictive ability of PKP3. Finally, we assessed the impact of PKP3 on the progression, migration, and invasion of PC cells.

## 2. Materials and Methods

### 2.1. Data Downloading and Processing

To date, The Cancer Genome Atlas (TCGA) and the Genotype-Tissue Expression (GTEx) contain a total of 178 PAAD tissues and 171 healthy pancreatic tissues. The University of California Santa Cruz (UCSC) dataset compiles and normalizes high-throughput sequencing (HTSeq) data from TCGA and GTEx, which can be utilized to conduct differential expression analyses of genes [21]. On 1 July 2022, we accessed the UCSC system and obtained HTSeq data for TCGA-PAAD, GTEx-pancreas, and TCGA Pan-Cancer, all formatted as transcripts per million. The clinical data pertaining to patients with PAAD were acquired from TCGA. Following the removal of specimens with incomplete survival data, 178 participants were retained for further investigation. To identify the presence of significant variations in the expression levels of PKP3 among the different subgroups, a Wilcoxon rank-sum test was carried out. We also screened the microarray expression data available in the Gene Expression Omnibus (GEO) for additional verification of the outcomes collected from the difference analysis [22].

### 2.2. PKP3-Related Functional Analysis

The co-expressed genes of PKP3 were retrieved from the TCGA-PAAD dataset to conduct Gene Ontology (GO) and Kyoto Encyclopedia of Genes and Genomes (KEGG) analyses. According to Spearman correlation analysis, the screening criteria of PKP3 co-expressed genes were set as the absolute value of the correlation coefficient ≥ 0.5 and *p* value < 0.05 [23]. Prior to conducting gene functional enrichment analysis, the gene symbols were converted to the EnterzID and GO and KEGG signaling pathway annotations, which were acquired with R software (R version 4.0.2, R Foundation, Vienna, Austria)). The clusterProfiler package was used to perform functional enrichment analysis of GO and KEGG (version 3.14.3).

### 2.3. Gene Set Enrichment Analysis (GSEA) of PKP3

The TCGA-PAAD specimens were segregated into two distinct groups, the PKP3-high group and the PKP3-low group, in accordance with the median expression value of PKP3. The GSEA software (4.0.2) was utilized to conduct a GSEA analysis comparing the PKP3-high and PKP3-low groups [24]. The dataset denoted as c2.cp.kegg.v7.4.symbols.gmt was employed as the control group, and the permutation number was established at 1000. Pathways that were significantly enriched had to match the established criteria, as follows: false discovery rate (FDR) less than 0.25, a normalized enrichment score absolute value (NES) greater than 1, and a nominal *p*-value (NOM *p*-value) less than 0.05.

### 2.4. Immune Infiltration Correlation Analysis

Using the ssGSEA algorithm, the connection between PKP3 expression and PC immune cell infiltration was investigated [25]. Using the GSVA R package, the constituents of 28 TICs in each sample of PC were obtained. Subsequently, a Wilcoxon test was employed to compare the 28 TICs’ infiltration levels between the PKP3-high expression group and PKP3-low expression group. The study employed Spearman correlation analysis to evaluate the relationship between the different expression levels of PKP3 and the composition of the 28 TICs. To better understand the potential mechanisms underlying the relationship between PKP3 and immune cell infiltration, we performed an evaluation of the connection between PKP3 expression and immune-associated genes, such as MHC genes, immune activation genes, immune activation genes, chemokines, and chemokine receptors.

### 2.5. Patients and Cell Lines

Immunohistochemistry analysis was performed on a total of 85 PC tissues and 45 healthy tissues. All specimens included in the study were confirmed to have pancreatic ductal adenocarcinoma through a postoperative pathological investigation. Prior to the surgical procedure, it was determined that none of the individuals underwent any prior radiotherapy or chemotherapy treatments. Before enrollment in the investigation, the participants were required to provide written informed consent. Additionally, the Ethics Committee of the First Hospital of Lanzhou University granted approval for all instructions pertaining to the use of human samples discussed in this study (No. LDYYLL2022-196). The PC cell lines, including ASPC-1, SW1990, BXPC3, and PANC-1, as well as the human pancreatic cell line (HPNE), were procured from the Shanghai Institute of Nutrition and Health (Shanghai, China).

### 2.6. RNA Extraction and qRT-PCR

RNA was extracted from the specimens utilizing TRIzol in qRT-PCR. Then, a PrimeScript RT Reagent Kit was used to generate cDNA. A StepOne Real-Time PCR Instrument (Applied Biosystems, New York, NY, USA) was utilized to conduct the qRT-PCR analyses. Next, the 2-ΔΔCt technique was utilized to assess the relative levels of gene expression, with GAPDH being used as the reference for normalization. Primers utilized in this research were as follows: forward, ACGTGCATGGGTTCAACAG; reverse, CTGCTGAGGGCAGGTCAAT. The assessments were performed in triplicate.

### 2.7. Cell Phenotypic Assay

The CCK-8 assay involved seeding cells in 96-well plates with 5 × 10^3^ cells per well at interval points of 0, 24, 48, and 72 h. CCK-8 reagents (GenePharma, Shanghai, China) were introduced into each well and subjected to an incubation period of 2 h at a temperature of 37 °C. The measurement of absorbance was conducted at 450 nm using a microplate reader (Thermo Fisher Scientific, Waltham, MA, USA). The cells were introduced into a 24-well Cell Culture Cluster to conduct a colony formation assay and evaluate wound healing. The cells were underlined perpendicular to the cell culture plate using a pipette tip. Scratch widths were measured at two interval points, 0 and 48 h, utilizing an optical microscope. In the context of transwell assays, the inserted upper chamber was seeded with cells at 2 × 10^4^ for the migration assay and 1 × 10^4^ for the invasion assay. A medium including 10% FBS was added to the lower chamber. The cell invasion assay involved pre-coating the upper chambers of a transwell device with diluted Matrigel solution (BD Biosciences, Franklin Lakes, NJ, USA). Following a twenty-four-hour incubation period, cells that underwent migration and invasion were subjected to fixation and staining procedures, followed by quantification and photographic documentation.

### 2.8. Immunohistochemical Assay

Immunohistochemistry was analyzed using a PKP3 antibody (Proteintech, Rosemont, IL, USA, diluted at 1:500). Two proficient pathologists conducted a semi-quantitative assessment of the immunohistochemical findings through visual observations. The proportion of positive cells (EXT) was categorized into one of four distinct grades: 0 (0%), 1 (<10%), 2 (10–50%), and 3 (>50%). Staining intensity (INT) was also divided into four grades: 0 (absence of staining), 1 (yellow), 2 (tan), and 3 (brown). The formula for calculating the relative expression index was as follows: (EXT*INT). Scores below 4 were considered indicative of poor expression, whereas scores equal to or greater than 4 were indicative of high expression.

### 2.9. Statistical Analysis

A statistical analysis of data was conducted using the R (version 4.0.2, R Foundation, Vienna, Austria), SPSS (version 25.0, IBM SPSS Inc., New York, NY, USA), and Prism (version 8.0.2, GraphPad Inc., California, CA, USA) software packages. The process of data visualization used the following instruments: the ggplot2 R package and Prism (version 8.0.2). Chi-square tests were used to examine variations within clinical features among groups expressing high and low levels of PKP3. A Spearman test was employed to assess the association between the expression of PKP3 and clinical features. The study also employed Kaplan–Meier survival curve analysis, as well as the univariate and multivariate Cox analysis approaches, to analyze the predictive significance of PKP3 within clinical specimens. *p* < 0.05 was considered statistically significant.

## 3. Results

### 3.1. PKP3 Relative mRNA Expression Was Found to Be Increased in PC

Initially, an examination was conducted on the expression of PKP3 mRNA in various types of cancers by utilizing a pan-cancer dataset obtained from TCGA. The findings showed that PKP3 mRNA was significantly elevated in 15 tumor types, including breast invasive carcinoma, lung adenocarcinoma, and pancreatic adenocarcinoma (Figure 1A). Nevertheless, patients diagnosed with kidney chromophobe, kidney renal clear cell carcinoma, and liver hepatocellular carcinoma exhibited a significant decrease in PKP3 mRNA expression (Figure 1A). PKP3 mRNA expression was heterogeneous among different tumor types but showed an increasing trend in most tumors, including PC. Next, a comparative analysis of the mRNA expression levels of PKP3 in PC tissues and normal pancreatic tissues was conducted utilizing data from the TCGA and GTEx datasets, which were further confirmed using microarray data from the GEO database. The results all showed the overexpression of PKP3 mRNA within PC tissues, in contrast with healthy pancreatic tissues (Figure 1B–G).

### 3.2. The Expression Levels of PKP3 mRNA Are Connected to the Clinical Characteristics

Based on the findings obtained from the TCGA database, we sought to further clarify the connection between PKP3 mRNA expression and clinical features in PC. After removing clinical information that could not be matched with expression data, the clinical features of 178 individuals were used for the analysis (Table 1). Patients were grouped according to their clinical characteristics, and the results of comparisons between groups demonstrated that a higher expression of PKP3 mRNA was correlated with gender (male vs. female, *p* < 0.01, Figure 1H), histologic grade (G3/4 and G2 vs. G1, *p* < 0.05, Figure 1I), survival state (dead vs. alive, *p* < 0.01, Figure 1J), and T stage (T3/4 vs. T2, *p* < 0.05, Figure 1K). Moreover, the univariate logistic regression analysis revealed that PKP3 mRNA expression was positively correlated with gender (male vs. female, OR = 2.085, *p* < 0.05), histologic grade (G3/4 vs. G1/2, OR = 2.152, *p* < 0.05), and survival state (dead vs. alive, OR = 1.887, *p* < 0.05, Table 2). The aforementioned findings indicate a significant association between overexpressed PKP3 mRNA and a negative tumor differentiation grade, as well as an adverse patient prognosis in the context of PC.

### 3.3. PKP3 Is a Potential Prognostic Marker in PC Individuals

The present investigation sought to analyze the prognostic significance of PKP3 in PC by exploring the survival outcomes of patients with varying levels of PKP3 mRNA expression on the TCGA dataset. The findings from the Kaplan–Meier survival analysis indicate that a significant correlation exists between elevated levels of PKP3 mRNA expression and reduced PFS, overall survival time, and disease-specific survival time (Figure 2A–C). We conducted a univariate and multivariate COX regression analysis to determine the independent predictive indicators that affect patient survival. The findings of the study demonstrated that patient survival was influenced by independent predictive variables including age, N stage, and PKP3 mRNA expression (Figure 2D). The prognostic significance of PKP3 was further evaluated through the construction of a nomogram. The nomogram model C-index result was 0.648 (95% CI: 0.583–0.713), demonstrating that the model offered a significant degree of prognostic efficacy (Figure 2E). The calibration plot represents the bias correction lines for the one-year and three-year intervals and a 45° diagonal line. This alignment indicates that the expected survival time closely corresponds to the theoretical value (Figure 2F). To summarize, PKP3 overexpression was strongly correlated with a negative patient prognosis and is a potential biomarker for PC.

### 3.4. PKP3 Protein Is Upregulated in PC and Connected with a Negative Prognosis

To verify whether the PKP3 protein translation levels were consistent with the mRNA transcript levels, we gathered 84 PC tissues and 40 healthy PC tissues to complete the immunohistochemistry analysis. The findings demonstrated that PKP3 expression also increased in PC tissues (49 cases showed high expression; 58.3%) but decreased in normal pancreatic tissues (26 cases showed low expression; 65%; Figure 3A). Using Cox regression and Kaplan–Meier survival curve analysis, we acquired additional confirmation of the prognostic significance of PKP3 at the protein level. The clinical features of 84 individuals diagnosed with PC are presented in Table 3. The survival of individuals with PC was found to be independently influenced by the expression of the PKP3 protein, as identified by both multivariate and univariate COX regression analyses. Survival was also correlated with age, T stage, and N stage (Figure 3B). The findings obtained from the Kaplan–Meier survival curve analysis demonstrated that the group with high PKP3 protein expression exhibited a significantly reduced survival time compared with the group featuring low PKP3 protein expression (Figure 3C). Consistent with the mRNA levels, PKP3 protein levels were also elevated throughout PC tissues and correlated with poor patient outcomes. In addition, PKP3 expression was evaluated in four cell lines of PC (AsPC-1, BxPC-3, PANC-1, and SW1990) and one normal cell line (HPDE) utilizing qRT-PCR. The findings indicated that PKP3 expression was elevated in the PC cell lines (Figure 3D).

### 3.5. Silencing PKP3 Inhibits PC Cell Progression

The sh-PKP3 plasmid was transfected into the SW1990 and AsPC-1 cell lines, and the qRT-PCR analysis results showed that PKP3 expression was significantly inhibited (Figure 4A). Subsequently, the aforementioned cells were utilized in CCK-8, wound healing, and transwell assays. In the CCK8 assay, the cellular proliferation rate exhibited a significant reduction in the PKP3-silenced group compared with the control group, suggesting that PKP3 promotes the proliferation of PC cells (Figure 4B,C). In the context of transwell assays and wound healing, the healing rate and number of cell penetrations in the PKP3-silenced group reflected poorer outcomes than those in the control group, indicating that PKP3 promotes the migration and invasive capacity of PC cells (Figure 4D–G). The aforementioned outcomes demonstrate that PKP3 can promote the malignant phenotype of PC cells, with the features of an oncogene.

### 3.6. The Function of PKP3 Is Involved in Tumor and Immune-Related Pathways

In accordance with the gene expression levels observed in TCGA pancreatic adenocarcinoma specimens, we screened PKP3 co-expressed genes among a number of 19,296 genes. Finally, a set of 1217 co-expressed genes was screened out with an absolute correlation coefficient >0.5 and a *p* value < 0.05. To better understand the prospective functions of PKP3 in PC, enrichment analyses of GO categories and KEGG pathways were conducted utilizing co-expressed genes. The outcomes indicated that PKP3-related GO categories were mainly involved in cell–cell junctions, cell morphology, and cell movement, consistent with the basic functions of the PKP protein family (Figure 5A). In addition, the biological function of PKP3 is closely related to the Rap1 signaling pathway, PI3K–Akt signaling pathway, and MAPK signaling pathway, illustrating the prospective function of PKP3 in promoting tumor progression (Figure 5B). GSEA was used to determine the regulatory role of PKP3 in the signaling pathways of a pancreatic malignancy. The enrichment findings showed that pathways connected with immune cells, immune diseases, and cytokines were enriched in the PKP3-low expression group, including natural killer cell-promoted cytotoxicity, the T-cell receptor signaling pathway, and the chemokine signaling pathway (Figure 5C). Taken together, PKP3 is a tumor-associated gene that may suppress immune responses against tumors.

### 3.7. PKP3 Inhibits TIC Infiltration and Immune Gene Expression in PC

To clarify the interaction between PKP3 and tumor immunology, the correlation between PKP3 expression levels and immunocyte infiltration in PC tissue specimens was investigated. The comparative analysis results indicate that compared to the low PKP3 expression group, there is a statistically significant decrease in the content of most TICs in the high PKP3 expression group (Figure 6A). Generally, PKP3 expression was negatively correlated with TIC infiltration according to the Spearman correlation test (Figure 6B). In contrast, the infiltration of NK CD56bright cells, NK CD56dim cells, and Th 17 cells was positively correlated with PKP3 expression (Figure 6B). Further investigations were conducted to examine the co-expression correlation between PKP3 and gene sets associated with the immune system. The findings indicated that a significant proportion of chemokine receptors, MHC genes, immune activation genes, and chemokines exhibited adverse associations with the expression of PKP3 (Figure 7A–D). In conclusion, upregulated PKP3 is strongly connected to low levels of immune cell infiltration and may be involved in immunosuppression within the microenvironment of PC.

## 4. Discussion

Plakophilins are mainly found within the desmosome and serve a crucial function in the constitution and functional preservation of the cellular desmosome [26]. The disruption of desmosome assembly, aberrant protein expression, or modification can result in pathological conditions characterized by the aberrant regulation of cellular proliferation and wound healing. Recently, several studies have documented increased levels of PKP3 in human cancers, indicating the possible contribution of PKP3 in the promotion of cancer. A study on breast cancer revealed that PKP3 exhibited higher expression levels in tumor tissues and showed a positive correlation with node positivity and histologic grade [20]. Research by Valladares et al. revealed that PKP3 mRNA expression levels were increased in the blood of patients with gastrointestinal cancers and were associated with cancer progression and risk of death [27]. Our results indicated consistently abnormal upregulation of PKP3 expression levels in PCs in the TCGA-PAAD dataset, paraffin sections, and cell lines. The outcomes of the survival analysis suggested that PKP3 is connected with negative outcomes in PC individuals and is an independent prognostic risk factor. In addition, PKP3 is involved in regulating malignant progression in PC patients, including lower levels of tumor differentiation and larger tumor volumes, which could partially explain the poor prognosis caused by PKP3.

The abnormal expression of PKP3 causes a loss of cell adhesion, which, in turn, produces highly active and invasive tumor cells, resulting in tumor metastasis. A recent study on keratinocytes revealed that the overexpression of PKP3 not only enhances ERK activity but also captures phosphorylated retinoblastoma in the cytoplasm, which promotes E2F1 activity and mediates the progression of the cell cycle. The aforementioned data elucidate the mechanism through which PKP3 mediates cellular proliferation and functions as an oncogene [28]. In addition, the upregulation of PKP3 in prostate cancer cells leads to an increased proliferation rate [12]. The findings indicate that PKP3 functions as an oncogene that promotes tumor progression. Our research suggests that PKP3 mediates the progression, migration, and invasive capabilities of PC cells, eventually leading to tumor progression and poor prognosis.

The results of the functional analysis revealed that PKP3 plays a complex role in PC, not only in the multiple oncogenic signaling pathways but also in relation to the immunosuppressive microenvironment. The findings of the KEGG enrichment analysis demonstrated that PKP3 performs a vital function in stimulating both the PI3K–Akt signaling pathway and the MAPK signaling pathway, which are essential factors for cell growth, differentiation, and the regulation of metabolism [29,30]. Several studies have documented that the PI3K–Akt signaling pathway and MAPK signaling pathway are dysregulated during PC development; specifically, activation of these two pathways can promote tumor cell proliferation, stimulate metabolic reprogramming, suppress autophagy and senescence, and contribute to cell migration and invasion [31,32]. Aberrant upregulation of PKP3 may be mediated by triggering the PI3K–Akt and MAPK signaling pathways to induce pancreatic cell malignancy and promote the malignant phenotypes of PC cells. In addition, immunoassays have shown that PKP3 inhibits the infiltration of multiple TICs and the expression of immune-related cytokines, including CD8+ T cells, a potent destructive immune effector cell group in the antitumor response that induces an immune clearance response [33]. However, several studies have shown that some genes in PC induce the formation of an immunosuppressive microenvironment by promoting chemokine secretion, aberrant tumor angiogenesis, and suppressive checkpoint activation, which ultimately impede the transport and function of CD8+ T cells [33,34,35]. Therefore, PKP3 may be involved in suppressing CD8+ T-cell infiltration and activation in the PC microenvironment. PC malignancies are considered to be immune-desert tumors characterized by a severe lack of infiltration of antitumor immune cells, particularly CD8+ T cells, in the tumor microenvironment. These tumors usually have low immunogenic mutant antigens, making it difficult for tumor patients to benefit from immunotherapy [36,37]. Recent studies have shown that the use of local immunosuppressants to enhance the infiltration levels of CD8+ T cells for these tumors represents a good therapeutic strategy [37]. Therefore, PKP3 has the potential to function as a viable therapeutic strategy for immunotherapy in the management of PC.

## 5. Conclusions

The findings of our investigation indicate that the expression of PKP3 is upregulated in PC and associated with a negative prognosis for individuals. Mechanistically, PKP3 is not only involved in the modulation of different tumor-associated signaling pathways but also extensively inhibits the PC immune infiltration level, particularly the immune function of CD8+ T cells. The abnormal upregulation of PKP3 within PC tissues promotes tumor cell proliferation and metastasis, inhibits antitumor immunity, and ultimately leads to poor prognosis. Therefore, PKP3 has been recognized as a potentially valuable prognostic factor for individuals diagnosed with PC, as well as a possible target for immunotherapy.

## Figures and Tables

**Figure 1 biomedicines-11-02472-f001:**
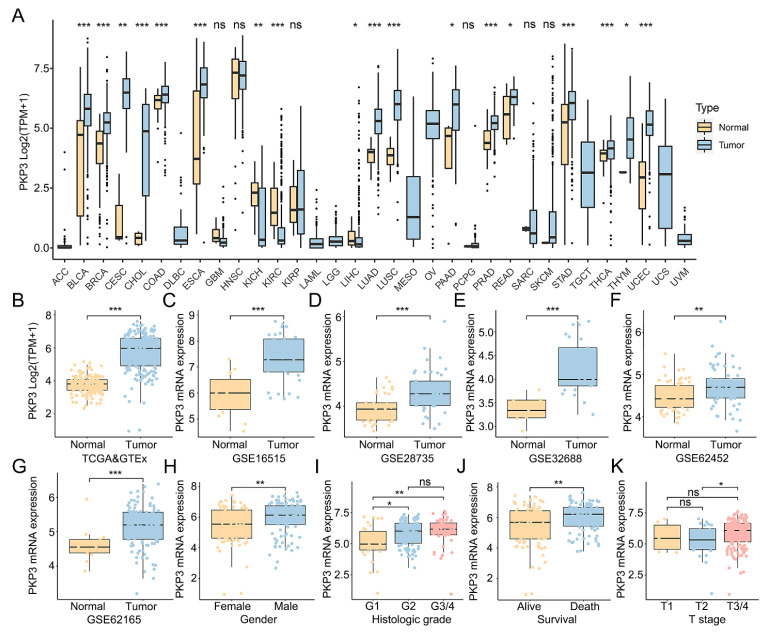
Expression levels of PKP3 mRNA in PC and their association with clinical characteristics. (**A**) Pan-cancer analysis of PKP3 expression in 33 malignant tumor types from the TCGA database. (**B**–**G**) Comparative analysis of PKP3 expression differences between PC tissues and normal tissues based on multiple research cohorts. (**H**–**K**) Comparison of PKP3 expression levels among PC patients grouped by different clinical characteristics. PKP3, plakophilin 3; PC, pancreatic cancer; TCGA, The Cancer Genome Atlas. *, *p* < 0.05; **, *p* < 0.01; ***, *p* < 0.001; ns, no significance.

**Figure 2 biomedicines-11-02472-f002:**
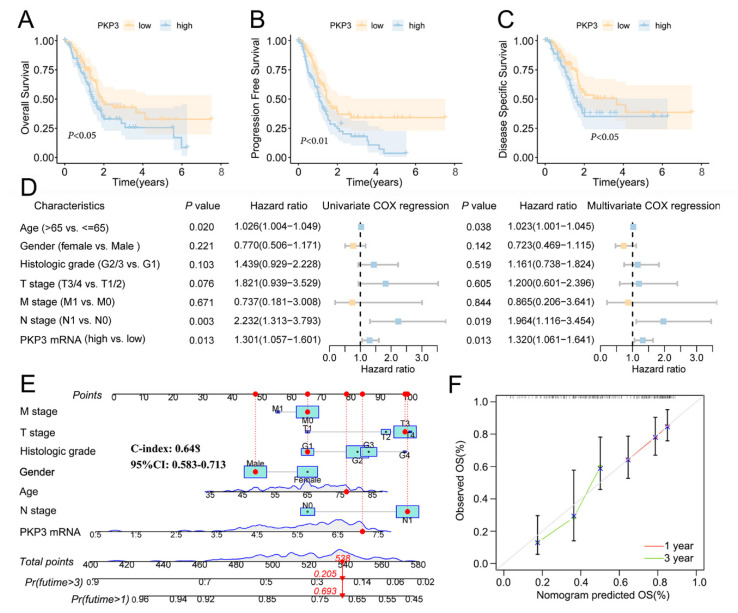
PKP3 mRNA shows a poor prognosis for PC individuals. (**A**–**C**) Kaplan–Meier survival curves of overall survival time, progression-free survival time, and disease-specific survival time; (**D**) univariate and multivariate Cox analysis of PKP3 mRNA and clinical features. (**E**,**F**) Nomogram model and calibration plot of PKP3 mRNA and clinical features. PKP3, plakophilin 3.

**Figure 3 biomedicines-11-02472-f003:**
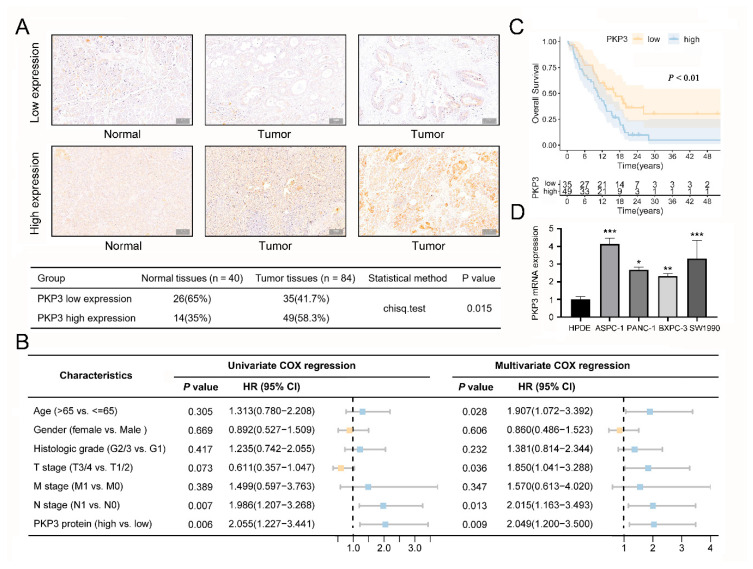
Validation of PKP3 expression differences at tissue and cellular levels in PC: (**A**) Immunohistochemical staining to determine the expression levels of PKP3 protein within PC tissues and normal pancreatic tissues; (**B**) univariate and multivariate Cox analysis of PKP3 protein and clinical features; (**C**) the Kaplan–Meier survival curve used to illustrate the prognosis of PC individuals with distinct PKP3 protein expression; (**D**) qRT-PCR analysis of PKP3 mRNA in the PC cell lines and normal pancreatic cell line. PKP3, plakophilin 3; PC, pancreatic cancer. *, *p* < 0.05; **, *p* < 0.01; ***, *p* < 0.001.

**Figure 4 biomedicines-11-02472-f004:**
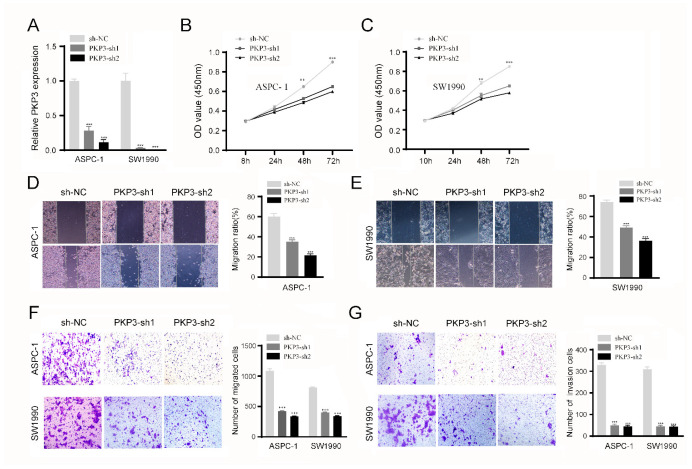
PKP3 knockdown suppresses the malignant phenotype of PC cells: (**A**) the qRT-PCR results used to determine the efficiency of PKP3 knockout in ASPC-1 and SW1990 cell lines; (**B**,**C**) the CCK-8 assay results used to analyze the impact of PKP3 knockout on tumor cell proliferation; (**D**–**F**) the wound healing assay and transwell migration assay results used to evaluate the impact of PKP3 knockout on tumor cell migration; (**G**) invasion assay results used to assess the effects of PKP3 knockout on tumor cell invasion. PKP3, plakophilin 3; PC, pancreatic cancer. **, *p* < 0.01; ***, *p* < 0.001.

**Figure 5 biomedicines-11-02472-f005:**
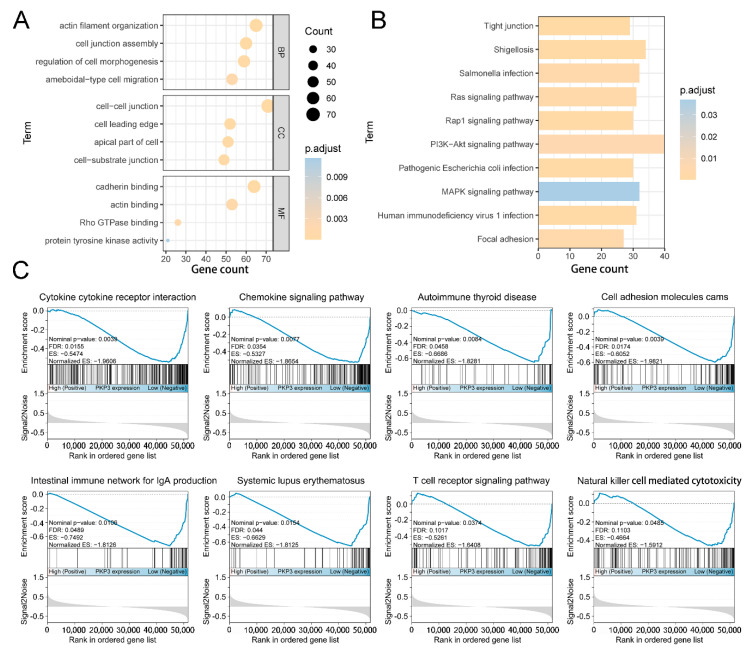
Comprehensive enrichment analysis associated with PKP3 in pancreatic cancer: (**A**) Gene Ontology (GO) analysis of PKP3 co-expressed genes; (**B**) Kyoto Encyclopedia of Genes and Genomes (KEGG) analysis based on PKP3 co-expressed genes; (**C**) immune-system-relevant signaling pathways and biological mechanisms in GSEA results. PKP3, plakophilin 3.

**Figure 6 biomedicines-11-02472-f006:**
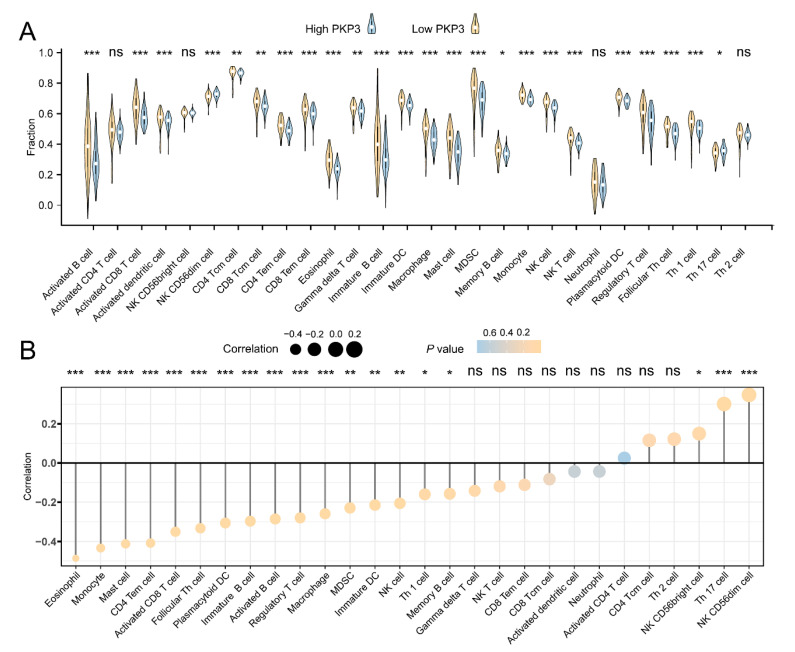
PKP3 expression is related to immune cell infiltration within PC: (**A**) Evaluation of the various levels of TIC infiltration in the high and low PKP3 expression groups. (**B**) Analysis of the correlation between the expression of PKP3 and the constituents of TICs. PKP3, plakophilin 3; TICs, tumor-infiltrating immune cells. *, *p* < 0.05; **, *p* < 0.01; ***, *p* < 0.001; ns, no significance.

**Figure 7 biomedicines-11-02472-f007:**
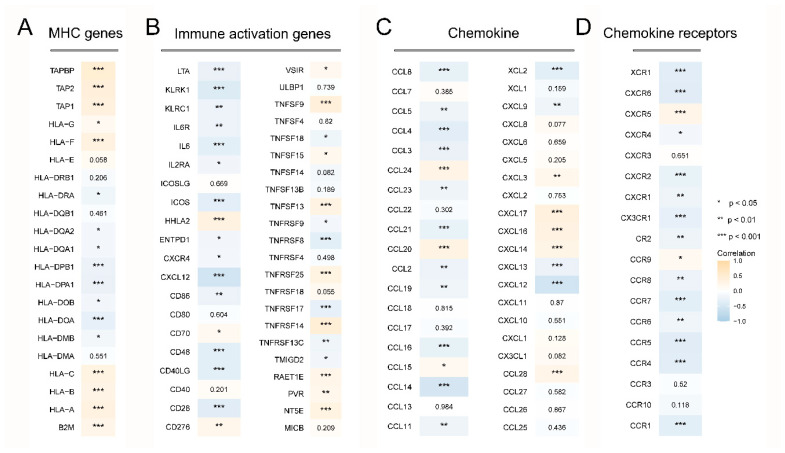
Association of PKP3 with immune-correlated gene groups: Association of EVL and MHC genes (**A**), immune activation genes (**B**), chemokines (**C**), and chemokine receptors (**D**). EVL, Ena-VASP-like. *, *p* < 0.05; **, *p* < 0.01; ***, *p* < 0.001.

**Table 1 biomedicines-11-02472-t001:** Clinical characteristics of pancreatic cancer patients in TCGA.

Characteristic	Levels	PKP3 mRNA Expression		*p* Value
		High	Low	
Age, n (%)	≤65	43 (48.31%)	50 (56.18%)	0.3679
	>65	46 (51.69%)	39 (43.82%)	
Gender, n (%)	Female	32 (35.96%)	48 (53.93%)	0.0238
	Male	57 (64.04%)	41 (46.07%)	
Chronic pancreatitis, n (%)	No	61 (85.92%)	67 (95.71%)	0.0855
	Yes	10 (14.08%)	3 (4.29%)	
Diabetes, n (%)	No	55 (73.33%)	53 (74.65%)	1
	Yes	20 (26.67%)	18 (25.35%)	
Histologic grade, n (%)	G1	8 (8.99%)	23 (26.44%)	0.0036
	G2	49 (55.06%)	46 (52.87%)	
	G3/4	32 (35.96%)	18 (20.69%)	
M stage, n (%)	M0	34 (89.47%)	45 (97.83%)	0.2513
	M1	4 (10.53%)	1 (2.17%)	
N stage, n (%)	N0	28 (31.82%)	22 (25.88%)	0.4881
	N1	60 (68.18%)	63 (74.12%)	
T stage, n (%)	T1	3 (3.37%)	4 (4.6%)	0.1635
	T2	8 (8.99%)	16 (18.39%)	
	T3/4	78 (87.64%)	67 (77.01%)	
Therapy outcome, n (%)	CR	26 (43.33%)	32 (61.54%)	0.0724
	PD	28 (46.67%)	12 (23.08%)	
	PR	3 (5%)	5 (9.62%)	
	SD	3 (5%)	3 (5.77%)	
Surgery type, n (%)	DP	16 (17.98%)	7 (7.87%)	0.0799
	Other Method	11 (12.36%)	8 (8.99%)	
	Whipple	62 (69.66%)	74 (83.15%)	
Pathologic stage, n (%)	Stage 1	7 (7.95%)	14 (16.09%)	0.215
	Stage 2	76 (86.36%)	70 (80.46%)	
	Stage 3/4	5 (5.68%)	3 (3.45%)	
OS state, n (%)	0	36 (40.45%)	50 (56.18%)	0.0512
	1	53 (59.55%)	39 (43.82%)	

**Table 2 biomedicines-11-02472-t002:** Univariate logistic regression assessment of the relationships between PKP3 and the clinical characteristics of pancreatic cancer.

Characteristics	Patients (n)	Odds Ratio (OR)	*p* Value
Survival state (dead vs. alive)	178	1.887 (1.041–3.423)	0.037
Gender (male vs. female)	178	2.085 (1.143–3.803)	0.017
Histologic grade (G3/4 vs. G1/2)	176	2.152 (1.095–4.23)	0.026
T stage (T3/4 vs. T1/2)	176	2.117 (0.946–4.734)	0.068

**Table 3 biomedicines-11-02472-t003:** Clinical characteristics of 84 pancreatic cancer patients.

Characteristics		PKP3 Expression		*p* Value
		Low, No. of Cases (n = 35)	High, No. of Cases (n = 49)	
Age	≤65	23 (65.7%)	36 (73.5%)	0.443
	>65	12 (34.3%)	13 (26.5%)	
Gender	Female	11 (31.4%)	14 (28.6%)	0.778
	Male	24 (68.6%)	35 (71.4%)	
Histologic grade	G1	10 (28.6%)	15 (30.6%)	0.84
	G2/3	25 (71.4%)	34 (69.4%)	
Pathologic stage	Stage1/2	29 (82.9%)	40 (81.6%)	0.885
	Stage3/4	6 (17.1%)	9 (18.4%)	
T stage	T1/2	27 (77.1%)	35 (71.4%)	0.557
	T3/4	8 (22.9%)	14 (28.6%)	
N stage	N0	18 (51.4%)	21 (42.9%)	0.437
	N1	17 (48.6%)	28 (57.1%)	
M stage	M0	33 (94.3%)	45 (91.8%)	0.667
	M1	2 (5.7%)	4 (8.2%)	
CA199	≤35	5 (14.3%)	13 (26.5%)	0.178
	>35	30 (85.7%)	36 (73.5%)	
CEA	≤5.2	28 (80%)	38 (77.6%)	0.787
	>5.2	7 (20%)	11 (22.4%)	

## Data Availability

The datasets presented in this study (sourced from TCGA, GTEx, and GEO) can be found in their respective online repositories. The datasets sourced from clinical samples are available from the corresponding author on reasonable request.

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
