# Peer review of "Comprehensive Analysis Identifies PKP3 Overexpression in Pancreatic Cancer Related to Unfavorable Prognosis"

_biomedicines, 2023, doi:10.3390/biomedicines11092472_

Round 1

Reviewer 1 Report

Dear colleagues, I’m very interested about your paper. The Pkp3 over expression is poorly studied but could be very interesting marker to understand prognosis of PC patients. In 2021 Liu published a small study about this with no results fo Pkp3 and in 2022 Deichmann not found on a large series of patients correlation between Pkp3 over expression and survival. Instead your paper well demonstrate this correlation. Could be a difference between different races? This could be a future idea for a larger and multicentrical study.

Author Response

Thanks to the reviewer for your appropriate and constructive comments, which made the revised manuscript far better than the previous one.

Point 1: Dear colleagues, I’m very interested about your paper. The Pkp3 over expression is poorly studied but could be very interesting marker to understand prognosis of PC patients. In 2021 Liu published a small study about this with no results fo Pkp3 and in 2022 Deichmann not found on a large series of patients correlation between Pkp3 over expression and survival. Instead your paper well demonstrate this correlation. Could be a difference between different races? This could be a future idea for a larger and multicentrical study.

Response 1: Thank you for your detailed feedback and the references provided. We have carefully read the papers you mentioned, and would like to address your comments as follows:

In the study by Liu et al., they performed survival analysis of PKP3 mRNA based on the TCGA-PAAD database. They found that the high expression group of PKP3 mRNA showed a trend of poor prognosis, but it was not statistically significant. In fact, we performed similar analysis in our research, but we obtained statistically significant differences. Upon rechecking the analytical process for this section, we are confident that our results are accurate. Moreover, the GEPIA database, an online TCGA analysis tool that has been cited over 1000 times, was used to analyze the TCGA-PAAD dataset. The results found that the high expression group of PKP3 mRNA was significantly associated with the prognosis of patients with pancreatic cancer (see attached figure).

In the study by Deichmann et al., PKP3 protein expression levels were detected by immunohistochemistry. They found that PKP3 expression was unrelated to the prognosis of patients with pancreatic cancer, which is different from our findings. Although the tissue samples originated from different racial groups, it is unlikely that racial factors have caused the discrepancy between the two studies. Our tissue samples yielded consistent results with the TCGA-PAAD data, which originated from the American population. We consider the heterogeneity of tissue samples to be the primary reason for this discrepancy, and we will increase the sample size in our subsequent research to verify the reliability of our results. Moreover, several findings in our research support the relevance of PKP3 to the prognosis of patients with pancreatic cancer:

  1. PKP3 is related to the differentiation grade and T stage of pancreatic cancer patients, and worse differentiation and tumor size may lead to poor prognosis;
  2. PKP3 can promote the proliferation, migration, and invasion of tumor cells, which may also result in a worse prognosis;
  3. Functional analysis of PKP3 suggests that PKP3 is involved in a variety of tumor-related signaling pathways, which may also be a potential mechanism by which PKP3 leads to poor prognosis.

We appreciate your constructive feedback and the opportunity to clarify these points.

Reviewer 2 Report

Comments and Suggestions for Authors

The manuscript by Du Y. et al investigated the potential prognostic value of plakophilins 3 (PKP3) as tumor biomarker for pancreatic cancer.

The authors found that high PKP3 levels were associated with poor clinical outcome, malignant biological properties of pancreatic cancer cells, and immune infiltration. The manuscript is well written, and the results are clearly described.

Some minor revisions are recommended.

Figure 4. The knockdown of PKP3 affected proliferation of the two cell lines ASPC-1 and SW1990 starting from 24 hours. The decreased proliferation can affect the result of then wound healing assay and transwell migration assay that are performed for 24 and 48 hours. The results of these assays should be adjusted for the proliferation rate of the two cell lines to clearly assess the effect of PKP3 on the migratory ability of cancer cells.

Paragraph 3.7 (lines 526-528). The authors stated that “… that the majority of TICs in the PKP3 high expression group exhibited a statistically significant increase in content in comparison to PKP3 low expression group (Figure 6A).” But actually, the high PKP3 group showed general decrease in immune cells.

Paragraph 3.7 (lines 534-537). In figure 6B the infiltration of most of immune cells showed negative correlation with PKP3 levels. However, NK and Th17 cells showed increased infiltration. NK cells can play an anti-tumor function and Th17 can have a pro-inflammatory activity. Also, classical and non-classical HLA-I and the costimulatory molecule CD70 are positively correlated with PKP3 levels (figure 7) Therefore, it cannot be referred as immunosuppressed tumor microenvironment. Further targeted investigations on immune cells function should be conducted.

Author Response

Thanks to the reviewer for your appropriate and constructive comments, which made the revised manuscript far better than the previous one.

Point 1: Figure 4. The knockdown of PKP3 affected proliferation of the two cell lines ASPC-1 and SW1990 starting from 24 hours. The decreased proliferation can affect the result of then wound healing assay and transwell migration assay that are performed for 24 and 48 hours. The results of these assays should be adjusted for the proliferation rate of the two cell lines to clearly assess the effect of PKP3 on the migratory ability of cancer cells.

Response 1:Thank you for your insightful comments. They are indeed crucial for the improvement of our study.

Regarding the CCK8 experiment, the results indicated a trend of decreased cell proliferation rate 24 hours after knocking down PKP3. However, the change was not statistically significant. To further investigate this, we conducted a scratch assay, choosing 0 hours and 24 hours as the time points.

We understand your concern about the time point selection. However, if we set the time points within 24 hours for the scratch assay, the results might not be clear due to the influence of cell migration rate. We believe that the current time points can provide a more accurate reflection of the effects of PKP3 knockdown on cell proliferation and migration.

We appreciate your valuable feedback and will continue to refine our experiments to ensure the validity and reliability of our results.

Point 2: Paragraph 3.7 (lines 526-528). The authors stated that “… that the majority of TICs in the PKP3 high expression group exhibited a statistically significant increase in content in comparison to PKP3 low expression group (Figure 6A).” But actually, the high PKP3 group showed general decrease in immune cells.

Response 2: Thank you very much for your comments. We apologize for the misrepresentation in our manuscript. We have now corrected the sentence in the manuscript to read, " The comparative analysis results indicate that, compared to the low PKP3 expression group, there is a statistically significant decrease in the content of most TICs in the high PKP3 expression group (Figure 6A)".

Point 3: Paragraph 3.7 (lines 534-537). In figure 6B the infiltration of most of immune cells showed negative correlation with PKP3 levels. However, NK and Th17 cells showed increased infiltration. NK cells can play an anti-tumor function and Th17 can have a pro-inflammatory activity. Also, classical and non-classical HLA-I and the costimulatory molecule CD70 are positively correlated with PKP3 levels (figure 7) Therefore, it cannot be referred as immunosuppressed tumor microenvironment. Further targeted investigations on immune cells function should be conducted.

Response 3: We appreciate your thorough evaluation of our work and would like to address the points raised in your review.In Figures 6 and 7, we aimed to assess the correlation between PKP3 expression and tumor-infiltrating immune cells (TICs) as well as immune gene sets. We acknowledge your observation that, despite a decrease in the infiltration levels of most TICs in the high PKP3 expression group, there is an elevation in the levels of NK cells and Th17 cells. Additionally, PKP3 expression appears to be positively correlated with certain immune-promoting factors. It is important to clarify our rationale for these findings. Our study employed bioinformatics methodologies with the intention of analyzing the overarching impact of PKP3 on the tumor immune microenvironment. Our results suggest that PKP3 inhibits the infiltration of the majority of immune cells as well as the expression of immune-related genes. However, we acknowledge the inherent limitations of bioinformatics analysis, and we concur that substantial experimental validation is required to fully comprehend the effect of PKP3 on tumor immunity.

Reviewer 3 Report

Thank you for submitting this interesting manuscript describing PKP3 expression as a prognostic biomarker in pancreatic cancer. However; I found it was very difficult to read this manuscript due to poor quality of English language and .  The design of the study and methodology were inappropriate.  There is no power of study given the small number of patients and heterogeneity of the patients cohorts.

Poor English language  with many grammatical errors.

Author Response

We appreciate your time and effort in reviewing our manuscript, Your feedback is invaluable to us in improving the quality of our work. We would like to express our sincere apologies for the difficulties you encountered while reading the manuscript due to the poor quality of English language and the inappropriate design of the study and methodology.

We have taken your comments seriously and have made significant revisions to address the issues you raised. In particular, we have enlisted the services of a professional editing agency with expertise in language refinement. They have conducted a comprehensive review of the manuscript and have made substantial improvements to the English language, enhancing the overall readability and clarity of the content.

Regarding the study's design and methodology, we acknowledge your concerns about the small number of patients and the heterogeneity of the patient cohorts. While the limited sample size presented certain challenges, we have re-evaluated our approach and have taken steps to provide a more comprehensive context for our findings. We have added a thorough discussion of the limitations arising from the patient cohort heterogeneity and have highlighted the need for further studies with larger patient populations to validate our results.

Once again, we thank you for your thoughtful comments and for guiding us toward making substantial improvements to our manuscript.

Round 2

Reviewer 3 Report

Thank you for submitting the revised manuscript that addressed my previous concerns. However the English language revision is still needed. For example the first sentence in the abstract “ The plakophilins 3 (PKP3) an affect cell signal transduction and cell adhesion and play an important role in ….” should be corrected.

Extensive English language editing is still required.

Author Response

Thank you for providing valuable feedback on our manuscript. We appreciate your insights and suggestions. In order to effectively address the language-related issues raised, we opted to utilize the professional English editing service offered by MDPI. The editing process has been completed under order number "english-70234". Building upon this, we conducted another thorough review of the manuscript's language and have confirmed the absence of any grammatical errors. Once again, we sincerely thank you for your time and attention to our work.